# Jordanian Kaolinite with TiO_2_ for Improving Solar Light Harvesting Used in Dye Removal

**DOI:** 10.3390/molecules28030989

**Published:** 2023-01-18

**Authors:** Manal Alkhabbas, Fadwa Odeh, Khitam Alzughoul, Razan Afaneh, Waed Alahmad

**Affiliations:** 1Department of Chemistry, Faculty of Science, Isra University, Amman 11622, Jordan; 2Department of Chemistry, The University of Jordan, Amman 11942, Jordan; 3Department of Geology, The University of Jordan, Amman 11942, Jordan; 4Department of Chemistry, Faculty of Arts and Science, Applied Science Private University, Amman 11931, Jordan

**Keywords:** sol-gel, kaolinite, photodegradation, nanocomposites, solar light harvesting

## Abstract

TiO_2_-Kaolinite nanocomposite photocatalysts were synthesized using the sol-gel method, with titanium isopropoxide/HCl as reactants and Jordanian kaolinite clay as a support material. The samples’ TiO_2_ content ranged from 10% to 70% (m/m). TiO_2_-Kaolinite composites were characterized using FTIR, SEM, XRF, and XRD. According to XRD measurements of the nano-composite samples, the intensity of the anatase peaks increased as the TiO_2_ percentage of the composition increased. As the percentage of TiO_2_ increased, so did the peaks of Ti-O-Si in FTIR. The extent of photocatalytic degradation of Congo-red dye was used to evaluate the photocatalytic activity of the prepared nanocomposites. After four hours under the sun, the percentage of Congo-red degradation ranged from 27 to 99 percent depending on the TiO_2_ content of the used nanocomposite. Meanwhile, the concentration drop in the dark did not exceed 10%. Photodegradation outperforms traditional treatment methods in terms of target degradation. Using naturally abundant materials such as clay in conjunction with metal oxides is widely regarded as an effective method of modifying the photoresponse properties of TiO_2_ particles, thereby improving solar light harvesting for target degradation.

## 1. Introduction

Many hundreds of millions of tons of organic pollutants (OPs), such as dyes, antibiotics, herbicides, and pesticides, are produced annually, which leads to water pollution problems [1]. Several technologies have been researched to treat wastewater contaminated by OPs, including biological degradation as well as chemical and physical methods [2]. The photodegradation of OPs using heterogeneous photocatalysts is an appealing method for wastewater treatment as it utilizes solar energy, it is inexpensive, and only mild conditions are required for the photodegradation reactions [3]. Several metal oxide photocatalysts, such as TiO_2_, ZnO, CeO_2_, etc., have been utilized for the degradation of OPs in the aqueous phase [4]. For example, carbon quantum dots decorated with MnIn_2_S_4_/Cu_2_O/Ag_2_O dual Z-scheme tandem hybrids were found to be very effective for dye degradation [5]. Titanium dioxide is widely studied as a photocatalyst due to its high photocatalytic activity. Furthermore, it is resistant to chemicals, insoluble in water, nontoxic, and inexpensive [6].

A pure photocatalyst has a small specific surface area, which results in poor adsorptive ability. To overcome this problem, researchers are trying to use support composites such as zeolite, mesoporous carbon, graphene, etc. [7]. Clays are considered interesting materials because they are abundant in nature, inexpensive, and have a high absorption and ion exchange potential. Due to their mineralogical composition, most clays are also suitable precursors for the synthesis of materials with improved properties [8,9,10].

The selection of suitable precursors for the adsorption of pollutants from aqueous solutions is critical for improving photodegradation. Recent advances in nanotechnology and engineering have proposed that nanomaterials can be used to purify water in a cost-effective manner while also having an efficient adsorption capacity for pollutants [11]. 

Several types of binders for the preparation of structured materials are described in the literature, including clays, alumina, and silica, as well as combinations of these materials. Due to their high mechanical and chemical stability, clays are a very promising alternative [12,13] for use in adsorption, catalysis, and the controlled release of active compounds. They also provide an intriguing way to revalue local natural resources. Because clays are inexpensive and widely available, they are an appealing binder for the immobilization of a wide range of active compounds in powder form. After the calcination process, the clays used as inorganic binders allow for mechanically resistant structured bodies. This thermally induced increase in mechanical strength is most likely due to the formation of new and stronger bonds between the constituent powders, which also improves the material’s wear resistance during the process [1,2,14].

As a result, using a heterogeneous photocatalysis to decontaminate or disinfect wastewater has gained prominence. Nonetheless, single-phase photocatalysts have limited catalytic activity due to their limited spectral activity, low quantum yield, and higher charge carrier recombination. Similarly, it is extremely difficult in a single catalytic system to include both the strong oxidation capacities of holes and the reduction capacities of electrons. These constraints have been efficiently organized through heterojunction construction using various photocatalysts with appropriate bandgaps for maximum optical absorption [5].

Jordan is a country abundant in clays of various origins. Industrial clays can be found in a variety of stratigraphic units ranging from the Paleozoic to the Cenozoic. The clay minerals with the greatest potential for industrial use are kaolinite, bentonite, and palygorskite. Other minerals found there include volkonskoite, illite/smectite, and smectite/kaolinite (Dead Sea black mud) [15,16,17].

This research investigates the addition of titanium oxide to Jordanian kaolinite in terms of diagnosis and application. It also measures the percentage of degradation of an organic pollutant model (Congo-red dye) to investigate the efficiency of prepared TiO_2_-Kaolinite photocatalysts.

## 2. Results and Discussion 

### 2.1. Characterization of TiO_2_-JK Nanocomposites 

Kaolinite [Al_2_Si_2_O_5_(OH)_4_] occurs in nature in association with illite, smectite, quartz, and heavy minerals. The structure of kaolinite is a combination of a silica tetrahedral sheet and a gibbsite sheet repeated indefinitely. Its particles are flaky and generally white, with a hexagonal outline, as seen in the SEM image (Figure 1). The disordered stacking of the sheets in disordered kaolinite is a result of shifting between the layers parallel to the b-axis. The presence of TiO_2_ nanoparticles (NP) mixed with kaolinite in various ratios is clear from the SEM images (60, 30 and 20% TiO_2_-JK NC. The size of the TiO_2_ NPs is about 10 nm, as evidenced from the SEM images. The finished products were in the form of a powder.

Pure kaolinite has a composition of Al_2_O_3_ (39.5%), SiO_2_ (46.5%), and structural water (14.0%). Above 400 °C, kaolinite loses the structural OH [15]. It is inert and can be used in industry over a pH range between 3 and 9 [17]. The most important commercial varieties are China clay, ball clay, refractory clay, and flint clay [17]. Kaolinite is used in various industries as paper filling and coating, ceramics, and fiberglass, as well as in cement, refractories, plastics, rubber, paint, geopolymers, cosmetics, polymers, and various catalysts and catalyst carriers. Modified raw kaolinite is used to create new support for the immobilization of metalloporphyrins [12,16,17,18]. Quantitative analysis of the TiO_2_-JK nanocomposites prepared in this study was performed using XRF. The compositions (listed in Table 1) show that the TiO_2_ percentage is close to the calculated ones after composite preparations.

The powder XRD pattern of JK used in this study is shown in Figure 2a, and it shows the characteristic peaks for kaolinite at 2θ of 12.3° and 24.8° [12,19]. Figure 2b represents the XRD for the prepared TiO_2_ NPs, which is consistent with anatase characterized by the peaks at 2θ of 25° [20]. As the percentage of TiO_2_ increases upon addition of TTIP in the prepared TiO_2_-JK, the characteristic peaks of kaolinite decrease and the characteristic peaks of anatase TiO_2_ increase but with a lower intensity because the amount of kaolinite is fixed (Figure 2).

The FTIR spectra of kaolinite, TiO_2_, and the different TiO_2_-JK NC were analyzed in the range of 400–4000 cm^−1^ and are shown in Figure 3. The peak at about 3600 cm^−1^ contributes to water molecules and surface hydroxyl groups. The band at about 1000 cm^−1^ corresponds to asymmetric stretching vibrations of SiO_2_ [21]. As the percentage of TiO_2_ increases, the intensity of this band decreases. The absorption band at 684 cm^−1^ in kaolinite could be attributed to Si-O bending; this band shifts to lower wave number due to the formation of Si-O-Ti, and its intensity decreases with increasing TiO_2_.

### 2.2. Photocatalytic Degradation Efficiency 

The TiO_2_-JK NC powders were prepared using the sol-gel method, which began with titanium tetraisopropoxide (TTIP) in a slightly acidic to neutral aqueous medium (pH 5.5–6.5). Following that, the nanoparticles were calcined for 5 h at 400 °C.

Velardi and others illustrated that the higher recombination of photo-generated electrons and holes of rutile with respect to anatase resulted in better photocatalysis by nanoparticles treated at 100 and 450 °C as compared to powder at 800 °C [22].

The 20 ppm concentration of CR and 0.1 g dosage of TiO_2_-JK NC were fixed in the experimental batch after pre-analysis to determine the optimized conditions, where $C_{0}$ is the initial CR concentration (mg. L^−1^) and $C_{t}$ is the concentration of CR at any time t. The CR degradation percentage (Removal efficiency) in each experiment was calculated as
(1)$$\%\;Degradation=\frac{C_{0}-C_{t}}{C_{0\;}}\times 100\%\;$$

Figure 4 shows the photodegradation of CR under sunlight irradiation. The absorption of the CR to calculate the degradation percentage was at 497 nm. According to the blank test (without catalyst), the concentration of CR does not change, indicating that CR is stable under solar irradiation. In addition, the CR concentration in the presence of kaolinite does not change after 30 min, whereas in the presence of TiO_2_-JK NC, its concentration decreases, indicating that the photocatalytic nanocomposite is successful in the degradation of CR (Figure 4). The ratios of 30 and 60% TiO_2_ provided the best photocatalytic activity for the degradation of CR under natural sunlight. Heterogeneous photocatalytic degradation occurs after the reactant has been adsorbed on the surface, adsorption being the precursor for the subsequent photocatalytic reaction [23]. 

The degradation efficiency of CR after 4 h under the selected parameters is shown in Figure 5. The highest percentages of degradation were found for 30 and 60% TiO_2_-JK NC and pure TiO_2_.

The degradation percentage for CR under dark conditions with continuous stirring ranged for the different ratios between 2 and 10%, proving the photodegradation process noticeably impacted absorption (Figure 6).

### 2.3. Reusability of TiO_2_-JK Photocatalyst

The reusability of the 30% and 60% TiO_2_-JK NC was investigated for four cycles of CR photodegradation. Figure 7 shows the reusability of the catalysts for four cycles after 4 h under solar irradiation. After the fourth cycle, the photo-activity decreases from 99.9% to 51.8% and from 99% to 40.1% for 30% TiO_2_-JK NC and 60% TiO_2_-JK NC, respectively. The decrease in the photocatalytic activity might be attributed to the loss of material during the washing and drying of the photocatalyst after each cycle. The active surface of the catalyst may also have changed during the four cycles due to aggregation [24].

## 3. Materials and Methods

### 3.1. Chemicals and Materials

Jordanian kaolinite (JK) used in this work was collected from Mahis and Hiswa areas. Both are attributed to the Kurnub Formation of the Lower Cretaceous age in Jordan.

Titanium isopropoxide (TTIP, 98%), ethanol (EtOH, 99.9), isopropanol (C_3_H_8_O), Congo-red (CR), and hydrochloric acid were purchased from Sigma-Aldrich (USA). CR was selected as the model compound to evaluate the efficiency of the titania-clay composites prepared as a photocatalyst for the degradation. Twice-distilled deionized water was used in all experiments. All chemicals were used without further treatment.

### 3.2. Synthesis and Characterization of TiO_2_-JK NC

The JK was first stirred at 400 rpm for 2 h at room temperature (25 ± 1 °C) at a 1% clay: distilled water ratio (*w*/*w*) (in order to remove unwanted soluble compounds). Then, a 0.4 M TTIP solution (solution A) was prepared by dropwise mixing 11.56 g TTIP, 0.913 mL HCl, and 87 mL ethanol; the pH of the solution was approximately 2.00.

Next, in a total volume of 200 mL (completed with Ethanol), solution B was prepared with various percentages of TTIP (70, 60, 50, 40, 30, 20, 10%) by taking the desired volume from solution A. Solution B was added dropwise to the previously prepared JK-water suspension and stirred for 24 h at 75 °C. After that, the mixture was filtered, washed with distilled water, and dried overnight in an oven at 100 °C. The solid powder was then calcined for 5 h at 400 °C. The TiO_2_ content in the TiO_2_-JK composites ranged from 10% to 70% *w*/*w*.

The chemical compositions of pure TiO_2_ as well as 20, 30, and 60% TiO_2_ in TiO_2_-JK composites were characterized by an X-ray fluorescence (XRF) spectrometer. The structural properties of kaolinite, TiO_2_, and TiO_2_-JK composites were examined by Powdered X-Ray Diffraction (PXRD), (7000 Shimadzu 2 kW model X-ray spectrophotometer instrument with a nickel filtered copper radiation (CuKa) with λ = 1.5456 Å, the 2θ range scan 2°–60° with 0.02° step size, Kyoto, Japan). The microstructures of the samples were examined using a scanning electronic microscope (SEM) [Hitachi High-Tech‘s scanning electron microscopes, Tokyo, Japan]. FTIR spectra of the samples were obtained over the 400–4000 cm^−1^ range using a spectrophotometer (Thermo Nicolet NEXUS 670, scan number was 32, the resolution was about 4.0 cm^−1^, using KBr pellets, Waltham, MA, USA).

### 3.3. Photodegradation of CR

Photodegradation experiments were conducted under direct sunlight from 10 am until 2 pm in September (2021) when the average temperature was 34 °C. Photodegradation processes were carried out by dispersing the TiO_2_-JK NP composite in a 100 mL of 20 ppm CR solution followed by stirring (optimized conditions). The experiments were conducted with different percentages of TiO_2_-JK composites. The photodegradation of the CR was examined by monitoring the decrease of CR concentrations at various time intervals by measuring the absorbance at the maximum absorbance wavelength (λ_max_ = 497 nm) of CR solutions using a UV-visible spectrophotometer (Varian Cary-100 UV/Vis spectrophotometer, Australia). The reusability of the 30% and 60% TiO_2_-JK composites were conducted for four cycles. The TiO_2_-JK NC batch (0.1 g of TiO_2_-JK in 100 mL of CR (20 ppm)) was kept for 6 h under sunlight, filtrated, then kept for the next day and used on another 100 mL of 20 ppm CR. This cycle was repeated for 5 days. At the end of each day, the concentration of CR in the filtrate was determined, and the TiO_2_-JK was characterized after the fifth day.

## 4. Conclusions

Jordanian kaolinite-supported catalysts (TiO_2_-JK NC) were prepared using the sol-gel method with different TiO_2_ percentages and were characterized using SEM, XRD, FTIR, and XRF: all techniques that are suitable for the preparation of TiO_2_-JK nanoparticles. 

The photocatalytic degradation technique has an advantage over traditional treatment methods; herein, we investigated the degradation percentage or the removal efficiency of CR from a body of water. The removal reached a high percentage in a short time (within 3 h). The use of naturally abundant materials such as clay in conjunction with metals is widely regarded as an efficient method of modifying the photoresponse properties of TiO_2_ particles.

More research is needed to apply photo-degradation or photoreactors based on modified nanocomposites in real-world polluted environments.

## Figures and Tables

**Figure 1 molecules-28-00989-f001:**
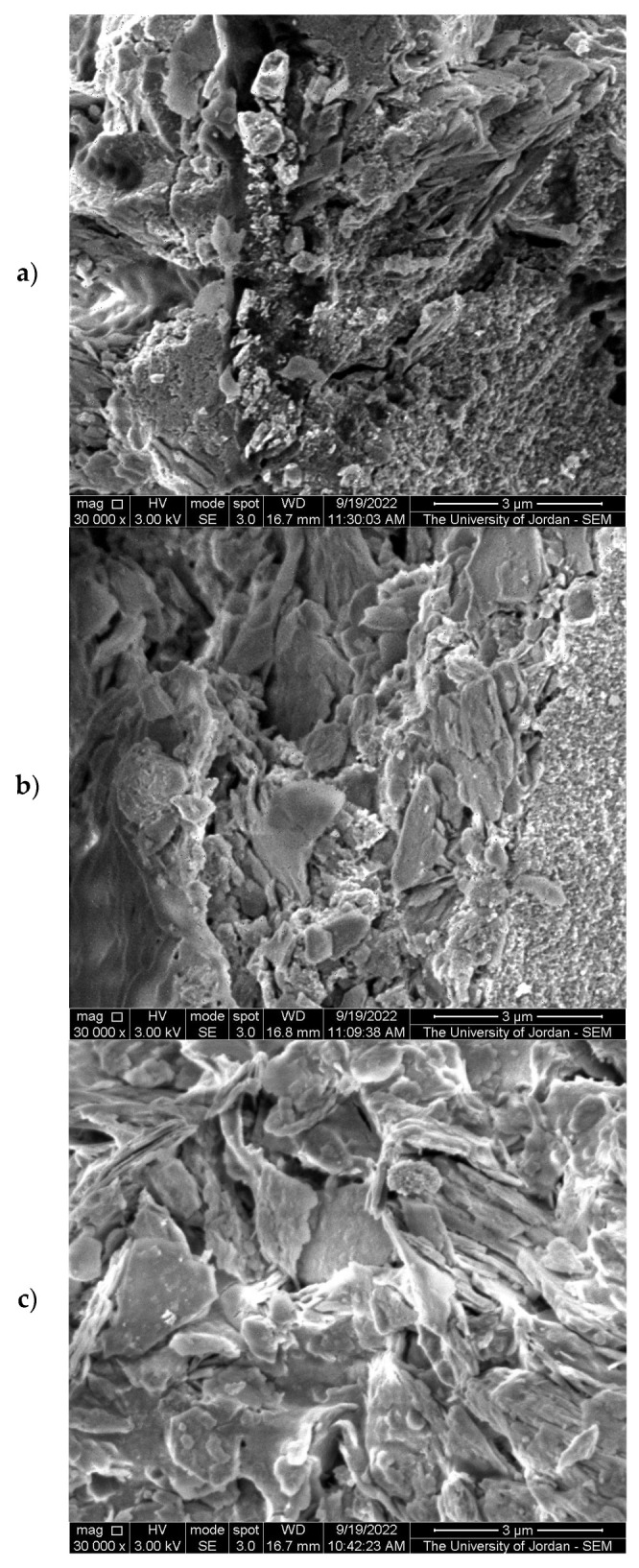
SEM images of (**a**) 60% TiO_2_-JK NC, (**b**) 30% TiO_2_-JK NC, and (**c**) 20% TiO_2_-JK NC.

**Figure 2 molecules-28-00989-f002:**
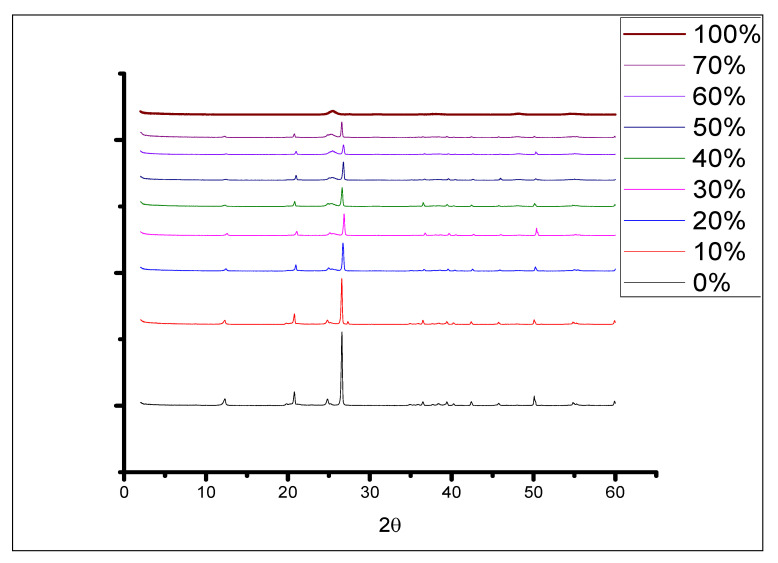
XRD patterns of the raw kaolinite and different ratios of TiO_2_ to kaolinite in TiO_2_-JK NC.

**Figure 3 molecules-28-00989-f003:**
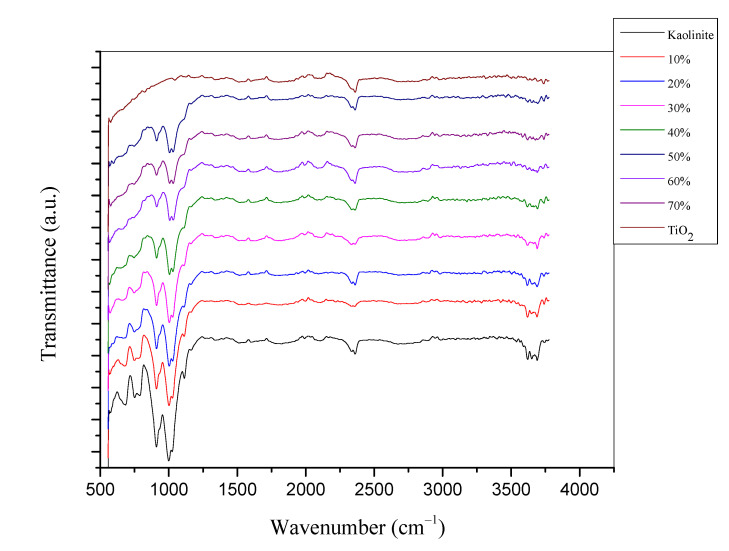
FTIR spectra of kaolinite, X% (X = 10, 20, 30, 40, 50, 60, 70): percentage of TiO_2_ in TiO_2_-JK NC.

**Figure 4 molecules-28-00989-f004:**
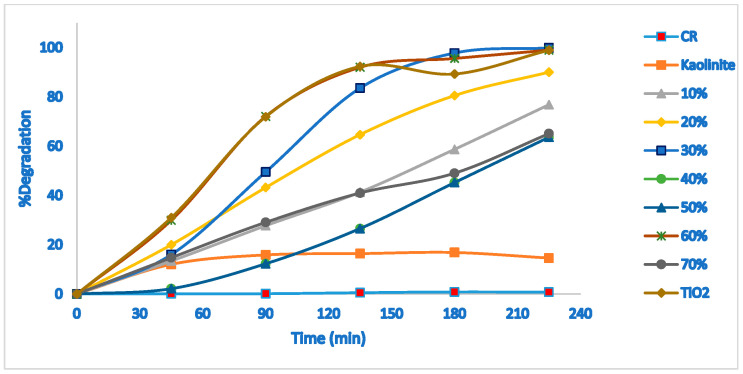
Photocatalytic degradation of CR under sunlight irradiation at various TiO_2_ percentages; CR, Kaolinite, X% (X = 10, 20, 30, 40, 50, 60, 70): percentage of TiO_2_ in TiO_2_-JK NC. Initial dye concentration is 20 ppm; photocatalyst dosage is 0.1 g.

**Figure 5 molecules-28-00989-f005:**
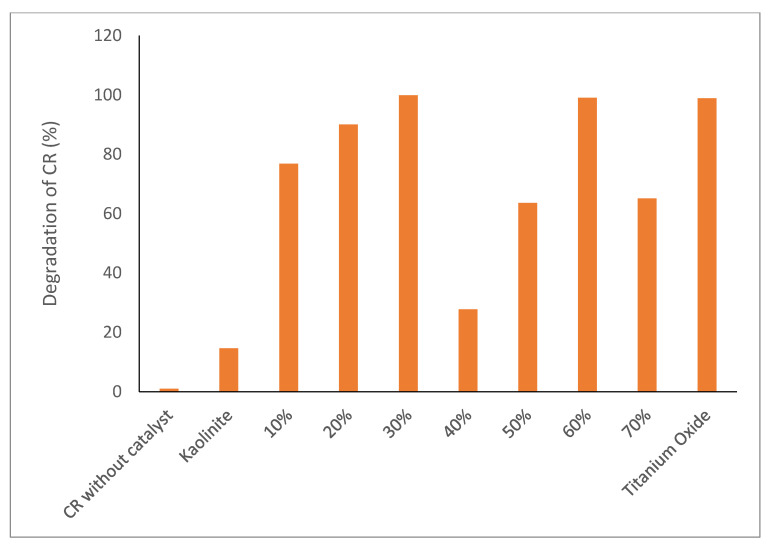
Photocatalytic degradation of CR in the kaolinite and in TiO_2_-JK.

**Figure 6 molecules-28-00989-f006:**
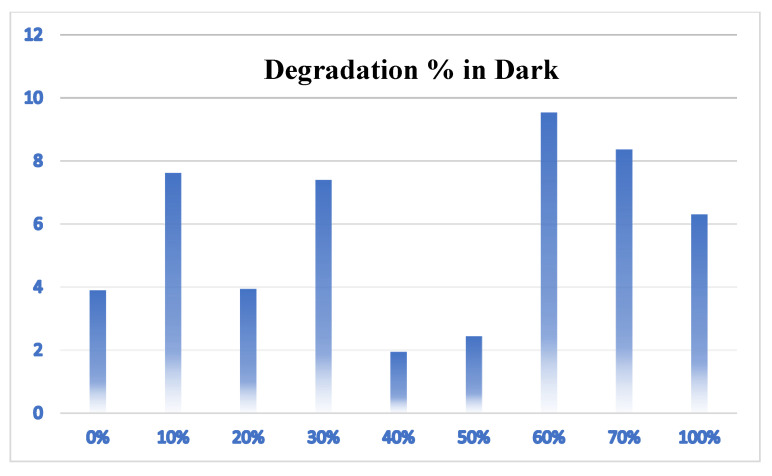
Photocatalytic degradation of CR under dark conditions.

**Figure 7 molecules-28-00989-f007:**
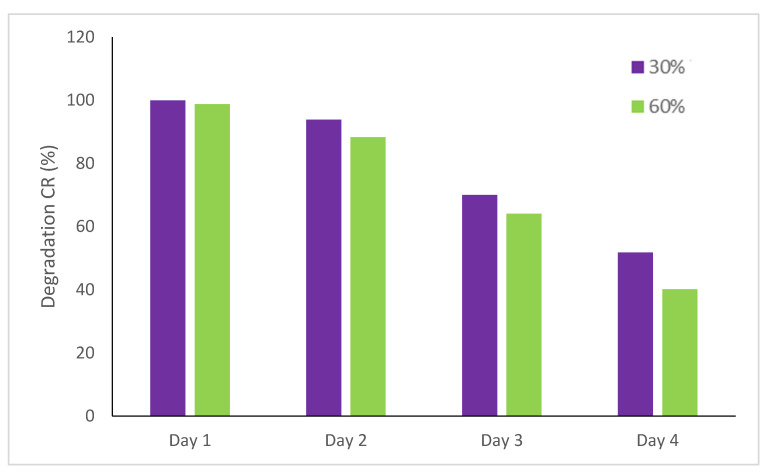
Reusability of 30% and 60% TiO_2_-JK NC for catalytic degradation after 4 h of solar irradiation.

**Table 1 molecules-28-00989-t001:** XRF data of the various TiO_2_-JK composites (as mass%).

%TiO_2_ in TiO_2_-JK NC	SiO_2_	Al_2_O_3_	TiO_2_	Fe
20	64.6	13.8	21.0	0.6
30	59.6	12.6	27.1	0.7
60	29.3	7.5	62.9	0.2
100	1.2	0.8	97.9	0.05

## Data Availability

The data presented in this study are available on request from the corresponding author.

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
