# Peer review of "Jordanian Kaolinite with TiO2 for Improving Solar Light Harvesting Used in Dye Removal"

_molecules, 2023, doi:10.3390/molecules28030989_

Round 1

Reviewer 1 Report

Reviewer’s comments:

The present article explores  the potential Jordanian Kaolinite with TiO2 for improving solar light harvesting used in dye removal Overall, the manuscript is well-prepared and organized. This topic is interesting and I recommend it to publish in this journal after minor revisions.

1.   Abstract should be rewritten based on the demand of the for the dye degradation  and how it is an ideal candidate.

2.   The authors can explain adsorption-coupled photoctalysis in pollutant  degradation. As for as semiconductor photocatalysts are concerned, adsorption plays a very critical role.  Semiconductor photocatalyst has as adsorption properties. Is degradation is affected by adsorption? The authors must add a paragraph about adsorption’s effect on heterogeneous photocatalysis.   The authors must discuss the basics of the heterogeneous photocatalytic process by discussing the latest work.  

         https://doi.org/10.1016/j.matlet.2022.131716

        https://doi.org/10.1016/j.surfin.2022.102182

3.    In addition to the literature presented on TiO2in the introduction, a few reports on bismuth-based and TiO2 may help the author in strengthening the introduction section to enhance the reader’s understanding

4.   Th authors need to  improve the resolution for more visuality. XRD figure is not clear.

5.   Author may highlight the methods/techniques that confirmed the formation of  photocatalyst.

6.   Conclusion is too conficonfinedit is suggested to write it in more detail.

7.     How the reusability of photocatalysts can be assured? magnetic adsorbents are of high importance.  How will we separate non-magnetic/metal-free photocatalysts How can we investigate catalytic activity over number of catalytic cycles?  Plz refer for some the papers for better explanation. What about the leaching of metals in solution? Plz add one paragraph regarding the separation of photocatalysts.

      https://doi.org/10.1016/j.jclepro.2017.04.085

Author Response

All corrections were done as recommended.

with regards  

Reviewer 2 Report

This study deals with the development of hybrid materials made of TiO2 and Jordanian Kaolinite. There are some characterizations. The English is not the best and could be revised by a native speaker, there are also a lot of typos (double space, double points, numbers that must be in subscript, etc). Also the layout of the paper is awful (some figures are cut, some tables are missing) and so it is difficult to follow the study. The paper needs major revisions before publication, here are some comments:

-          The scales in Fig 1 are quite small and can be increased in size.

-          The Table 1 is empty.

-          Fig 2 cannot be read.

-          No legend for the color in Fig 3.

-          Layout of Fig 4 is awful.

-          A comparison with literature is missing concerning hybrid TiO2-kaolinite materials (exemple: Mahy, J.G.; Tsaffo Mbognou, M.H.; Léonard, C.; Fagel, N.; Woumfo, E.D.; Lambert, S.D. Natural Clay Modified with ZnO/TiO2 to Enhance Pollutant Removal from Water. Catalysts 2022, 12, 148. https://doi.org/10.3390/catal12020148à

Author Response

(The authors gave the same response as above.)

Reviewer 3 Report

Journal: Molecules (ISSN 1420-3049)

Manuscript ID: molecules-2119377

Type: Article

Title: Jordanian Kaolinite with TiO2 for improving solar light harvesting used in dye removal.

Authors: Manal Alkhabbas , Fadwa Odeh * , Khitam Alzughoul , Razan Afaneh , Waed Alahmad *.

a)           Introduction: Write the objective of the present work carefully.

b)          How the author(s) measure the grain size of the TiO2 NPs and Kaolinite particles, explain?

c)           Table 1 is missing

d)          Figure 2 is missing too

e)           What is the type of samples that the author prepared powder or pellet?

f)            What is the temperature degree that the author used to do the composite 400oC? Is this temperature is enough for TiO2?

g)           Fig. 5 it’s better to use curves instead of points (photocalatytic degradation).

h)          Why the authors didn’t measure the other mechanical properties of the samples such as wear, impact, surface roughness, hardness (shore D), thermal conductivity, flexural strength….etc.?

i)             What does it add to the subject area compared with other published material?

j)             Paragraph 3, 3.1, 3.2, 3.3, and 3.4 should be written before paragraph 2 (results and discussion), so revise this in the manuscript.

k)          What is the range of wavelengths was used for UV-Visible spectrophotometer?

l)             For references, choose recent refs. Please, refer to these refs. are very useful for the photocatalytic activity

DOI:  https://doi.org/10.1016/j.optmat.2022.112725

DOI:  https://doi.org/10.1007/s43207-022-00254-5

Best Regards

Author Response

(The authors gave the same response as above.)

Round 2

Reviewer 1 Report

The revisions are OK.

Reviewer 2 Report

Corrections were done